# Introducing Fractal Dimension for Interlaminar Shear and Tensile Strength Assessment of Mechanically Interlocked Polymer–Metal Interfaces

**DOI:** 10.3390/ma13092171

**Published:** 2020-05-08

**Authors:** Erik Saborowski, Philipp Steinert, Axel Dittes, Thomas Lindner, Andreas Schubert, Thomas Lampke

**Affiliations:** 1Materials and Surface Engineering Group, Faculty of Mechanical Engineering, Chemnitz University of Technology, Erfenschlager Straße 73, D-09125 Chemnitz, Germany; axel.dittes@mb.tu-chemnitz.de (A.D.); th.lindner@mb.tu-chemnitz.de (T.L.); thomas.lampke@mb.tu-chemnitz.de (T.L.); 2Micromanufacturing Technology Group, Faculty of Mechanical Engineering, Chemnitz University of Technology, Reichenhainer Straße 70, D-09126 Chemnitz, Germany; philipp.steinert@mb.tu-chemnitz.de (P.S.); andreas.schubert@mb.tu-chemnitz.de (A.S.)

**Keywords:** fractal geometry, interlaminar tensile strength, interlaminar shear strength, strength prediction, roughness evaluation, mechanical interlocking, surface structuring, polymer–metal hybrid, laser micromachining

## Abstract

The interlaminar strength of mechanically interlocked polymer–metal interfaces is strongly dependent on the surface structure of the metal component. Therefore, this contribution assesses the suitability of the fractal dimension for quantification of the surface structure, as well as interlaminar strength prediction of aluminum/polyamide 6 polymer–metal hybrids. Seven different surface structures, manufactured by mechanical blasting, combined mechanical blasting and etching, thermal spraying, and laser ablation, are investigated. The experiments are carried out on a butt-bonded hollow cylinder testing method that allows shear and tensile strength determination with one specific specimen geometry. The fractal dimension of the metal surfaces is derived from cross-sectional images. For comparison, the surface roughness slope is determined and related to the interlaminar strength. Finally, a fracture analysis is conducted. For the investigated material combination, the experimental results indicate that the fractal dimension is an appropriate measure for predicting the interlaminar strength.

## 1. Introduction

Thermoplastic polymer–metal hybrids (PMH) offer great potential for lightweight applications owing to a high strength/stiffness-to-weight ratio and easy processability. In this context, thermal joining is a widely used process for creating adhesion between both dissimilar material groups. The thermoplastic polymer itself is used as hot melt adhesive as it infiltrates and interlocks the microstructural features of the metal surface. The achievable interlaminar strength depends in particular on the material pairing, the joining technique, as well as the surface structure of the metal component.

The material pairing determines the specific adhesion mechanisms that contribute to the interlaminar strength. Depending on the polymer, various adhesion mechanisms can occur. For example, for polyamide, Amend et al. [1] report dipole interactions, hydrogen bonds, and dispersion forces between amino, methyl, and carbonyl groups of the polymer and oxides, as well as hydroxides of the metal surface.

The joining technique determines the time–temperature–pressure regime. According to Zopp et al. [2], the cooling rate influences the crystallization and the related mechanical properties of the polymer within the melting zone. Katayama and Kawahito [3] investigated direct laser heating of the interface zone using transparent polymer. Bergmann and Stambke [4] used indirect laser heating of the metal component. Mitschang et al. [5] presented an inductive joining process. Liu et al. [6] presented a friction lap welding process, where a rotational tool is used to generate frictional heat on the metal component. Haberstroh and Sickert [7] used direct heat conduction to the metal component. Wagner et al. [8] presented an ultrasonic joining process, where frictional heat is generated directly between metal and polymer.

Moreover, the metal surface structure has a major impact on the achievable adhesion. Consequently, a large number of studies are focused on the relation between the surface structure and the interlaminar strength. Pan et al. [9], Bergmann and Stambke [4], as well as Saborowski et al. [10] investigated corundum blasting with various abrasive particle sizes. Their results indicate that larger particle sizes slightly increase the interlaminar strength. Amend et al. [11] presented various laser generated grid and crater-like microstructures, concluding that the interlaminar strength is related to the deepness of the structures. Steinert et al. [12] presented self-organized pin structures generated by a defocused laser beam, where densely arranged pin elements with a high aspect ratio arise from the surface. The interlaminar strength achieved with these pin structures could exceed the strength of other presented micro structures by far. Lindner et al. [13] achieved considerable adhesion with a nickel/aluminum thermal spray coating, comparable to a laser generated grid structure. The coating offered a broad variety of structure sizes as well as undercuts that were presumably beneficial for the interlaminar strength. Kleffel and Drummer [14] presented an electrochemical etching process, where a combination of nitric and hydrochloric acid leads to a massively undercut surface structure and a very high interlaminar tensile strength. Considering the mentioned studies, the following surface properties seem to have a positive effect on the interlaminar strength:A high structure density characterized by densely arranged profile elements with a high aspect ratio;The presence of sub-structures on the profile elements;The presence of undercuts.

However, an appropriate measure considering all these properties and allowing predictions of the interlaminar strength has not yet been presented for PMH. Kleffel and Drummer [14] as well as Bergmann and Stambke [4] found no considerable correlation between the interlaminar strength and standardized roughness height parameters like *Ra*, *Rz*, and *Rc*. Chen et al. [15] assessed the correlation between several roughness parameters and the shear strength of a steel–bone cement joint. A good accordance was found for the root mean square slope *R∆q*. This measure is directly connected to the structure density, as the slope depends on the height of the profile elements in relation to their distance. Saborowski et al. [10] verified these findings on aluminum/polyamide 6 (PA6) PMH pretreated with different surface structuring methods. However, undercuts could not be considered with this approach. Additionally, a considerable loss of detail occurs for small and dense surface structures owing to insufficient penetration of the profile elements with the stylus profiler tip. Thus, sub-structures are only considered partially. Amada and Yamada [16] introduced and successfully applied the fractal dimension for adhesive strength evaluation of plasma-sprayed ceramic coatings. The interface line was deduced from cross-sectional images and analyzed with the box-counting algorithm. Thereby, the surface structure is characterized with a clear numerical value. Because a high structure density as well as the presence of undercuts and sub-structures increase the fractal dimension, a direct relation between this measure and the interlaminar strength can be assumed.

The aim of this contribution is to adapt Amada’s and Yamada’s approach [16] for assessing the interlaminar strength of EN AW-6082/PA6 PMH. Therefore, the fractal dimension of seven differently structured surfaces is determined and correlated to the interlaminar shear and tensile strength. Four out of seven surface structures and the corresponding interlaminar strength values are deduced from previous investigations conducted by Saborowski et al. [10]. Three additional surface structures are created by laser ablation processing. The fractal dimension is determined by applying the box-counting algorithm on scanning electron microscopy cross-sectional images. In addition, the surface roughness slope is determined and evaluated. The specimens are joined by heat conduction hot pressing. Interlaminar strength values are determined using butt-bonded hollow cylinder specimens. This approach was initially presented by Mahnken and Schlimmer [17] for adhesive testing. Saborowski et al. [18,19] adapted the testing method for PMH, as it allows interlaminar shear and tensile strength testing with one single specimen geometry. Finally, the fracture surfaces are characterized.

## 2. Materials and Methods

### 2.1. Materials

For all experiments, hollow cylinders made up from EN AW-6082 aluminum alloy were used as the metal part. Hollow cylinders made up from extruded Ultramid^®^ B3 PA6 (BASF, Ludwigshafen, Germany) were used as the polymer part. The relevant material data are shown in Table 1. The parameters shown for the PA6 are given for the humid condition. The moisture content was adjusted by conditioning the material according to ISO 1110 (343 K/62% humidity). The conditioning of the PA6 took place before testing. In addition to that, the PA6 was dried before joining (343 K/ambient humidity). Thereby, the formation of cavities from evaporating water inside the melting zone is avoided.

### 2.2. Surface Pretreatment

#### 2.2.1. Grit Blasting

WFA F16 (particle size: 1000–1400 µm) corundum particles (F16) were applied to the aluminum with a blasting distance of 100 mm, a blasting angle of 75°, a blasting pressure of 0.2 MPa, and a treatment time of 10 s. In addition to the blasted-only structures, alkaline etching was conducted on grit blasted surfaces (F16-NaOH) to add much finer structural features to the comparably rough blasted surface. Thereby, a blasting pressure of 0.3 MPa was applied to increase the vertical extent of the structures. Alkaline etching was conducted with 2% NaOH solution (343 K/5 min). Afterwards, the sheets were submerged into 50% HNO_3_ solution (ambient temperature/2 min) in order to clean the surface.

#### 2.2.2. Thermal Spraying

A nickel-aluminum 95–5% coating (NiAl5) was deposited on the aluminum to create a rough and undercut surface structure. Prior to the coating process, WFA F24 (particle size: 600–850 µm) corundum particles were applied with a blasting distance of 100 mm, a blasting angle of 75°, a blasting pressure of 0.2 MPa, and a treatment time of 10 s to enhance the adhesion of the coating. The coating was applied by electric wire arc spraying, using a VISU ARC 350 arc spray system with Schub 5 spraying gun (Oerlikon Metco, Wohlen, Switzerland) with a current of 150 A, a voltage of 30 V, a spraying distance of 130 mm, an air pressure of 3.5 bar, a feed speed of 0.6 m/s, and a row spacing of 5 mm.

#### 2.2.3. Laser Structuring

The laser structuring processes were conducted by a Nd/YVO_4_ nanosecond laser system (specifications: wavelength = 532 nm, pulse duration = 10 ns, max mean power = 13 W, focus diameter = 15 µm; manufacturer: Spectra Physics^®^, Santa Clara, CA, USA). Multiple, line-wise scanning of the specimen’s surface area with overlapping single pulses was performed for the realization of the surface micro structures.

Stochastically distributed pin microstructures (L-Pin) were created in accordance with the work of Baburaj et al. [20] and Steinert et al. [12]. Thereby, a defined energy input above the material-specific threshold laser fluence is applied to the aluminum surface. A laser intensity of 3–6 J/cm² was realized using a defocused laser spot measuring 55 µm in diameter.

Deterministically distributed profile elements were generated with a focused laser beam; a pulse frequency of 200 kHz; and a number of 8, 11, and 14 scans (L-8, L-11, L-14). The resulting material ablations on the aluminum surface measure approximately 14 µm in diameter. By setting the distance between the material ablations slightly below their diameter (13 µm), the material in between is perforated. The resulting surface structure is determined by pin-like profile elements distributed in a grid. Thereby, the number of scans adjusts the height of the pins.

### 2.3. Butt-Bonded Hollow Cylinder Specimens

Figure 1a shows the geometry of the hollow cylinder specimens. The outer diameter *d*_o_ = 28 mm and the inner diameter *d*_i_ = 23 mm. The length of the metal cylinder *l*_m_ = 40 mm and the length of the polymer cylinder *l*_p_ = 60 mm. For metal cylinder, the free testing length *l*_m,f_ = 20 mm. For the polymer cylinder, the free testing length *l*_p,f_ = 30 mm. In case of the laser-structured specimens, *l*_p,f_ was reduced to 10 mm because the maximum twist angle of the testing machine (90°) would have been exceeded otherwise.

The specimens were manufactured using a heat conduction hot-pressing process. Figure 1b illustrates the hot-pressing tool. Beforehand, metal and polymer components were ultrasonically cleaned in ethanol. During the hot pressing process, an isobaric pressure (0.2 MPa) was applied to the specimen interface. The copper block was heated until the polymer in the interface melted and then air-cooled until the interface temperature reached 373 K. Finally, the pressure was removed and the specimen was taken. The resulting joining time was approximately 10 min per specimen. After the joining process, the specimens were reworked by turning to ensure equal measurements as well as enough centricity for testing. A detailed description of the manufacturing process can be found in a preceding study of Saborowski et al. [19].

The strength testing was conducted with a PTT 250 K1 hydraulic testing machine (Carl Schenck AG, Darmstadt, Germany). The specimens were clamped with ER40-472E collets according to ISO 15488. A steel plug was inserted into the polymer cylinder in order to prevent it from yielding when the collet was tightened. For determining the interlaminar shear strength *τ*_max_, the specimens were twisted until fracture. *τ*_max_ was calculated from the maximum torque *T*_max_ divided by the polar section modulus *W*_p_. For determining the interlaminar tensile strength *σ*_max_, the specimens were pulled until fracture. *σ*_max_ was calculated from maximum tensile force *F*_max_ divided by the overlapping area *A*_o_.
(1)$$\tau_{\max}=\;\frac{T_{\max}}{W_{P}}=\;\frac{{16T}_{\max}d_{o}}{{\pi (d}_{o}{}^{4}-d_{i}{}^{4})}$$
(2)$$\sigma_{\max}=\frac{F_{\max}}{A_{o}}=\frac{{4F}_{\max}}{\pi \left(d_{o}{}^{2}-d_{i}{}^{2}\right)}$$

The testing speeds were determined from the strain- and shear-rate of the PA6, whereby the much stiffer aluminum was considered rigid. The shear rate was set to 0.002 1/s (laser-structured: 5°/min, others: 15°/min) and the strain rate was set to 0.0002 1/s (laser-structured: 0.12 mm/min, others: 0.36 mm/min).

### 2.4. Fractal Dimension

The term fractal refers to the work of Mandelbrot [21]. Fractal geometry permits non-integer dimensions that describe the complexity of natural objects such as rough surfaces. In this context, Figure 2 illustrates how the interface line is deduced from cross-sectional images and how the fractal dimension approach is applied to it. The cross-sectional image (2048 by 1536 pixels) is recorded with a LEO1455VP scanning electron microscope (Carl Zeiss Microscopy GmbH, Jena, Germany). The image is further processed and binarized with GIMP 2 image manipulation software. Thereby, embedding resin and metal surface are separated from each other by the fuzzy select function that allows area selection based on color similarity. Afterwards, a MATLAB^®^ algorithm is used for the identification of the interface line. The identified line is transferred and centered to a white, quadratic image, whereby the side length in pixels equals a power of 2 (e.g., 2048 pixels). Finally, the box-counting algorithm is applied. Thereby, the image is divided in squares of size *r*. For each *r*_i_, a certain number of squares *n*_i_, containing at least one black pixel, exists. Beginning from the smallest possible box size of *r*_min_ = 1 pixel, the box size is increased stepwise by a power of 2 until the maximum box size *r*_max_, covering the complete image, is reached. Plotting n(r) in a logarithmic scale results in Figure 3a. The average negative slope of this curve equals the fractal dimension of the interface line. The slope between individual box sizes *d* shown in Figure 3b is calculated by Equation (3). The fractal dimension of the interface line *D* is calculated according to Equation (4).
(3)$$d_{i}{=\log}_{2}n_{i+1}-\log_{2}n_{i}$$
(4)$$D=\;\frac{1}{k}\overset{k}{\underset{i=1}{\sum }}d_{i}\;\mathrm{with}\;k=\;\log_{2}r_{\max}$$

Figure 4 and Figure 5 illustrate two theoretical examples, showing how characteristics beneficial for mechanical interlocking adhesion (structure density, amount of undercut surface area, and sub-structures) affect the value of *D*. In both cases, virtual cross-sectional images were created and evaluated with the algorithm illustrated in Figure 2. Figure 4a shows a dovetail structure. The structure density and the amount of undercut surface area are determined by the height-to-width ratio *h/w* and undercut angle *α*, respectively. Figure 4b shows how *D* increases with ascending *h/w* as well as *α*.

Figure 5 shows the first four iteration steps of the Koch curve, which is a commonly used example for fractal geometry. After the first iteration step, the total track consists of four straight sections of equal length, arranged at an angle of 0°–60°–120°–0°. In each further iteration step, each straight section is replaced by the total track of iteration step 1 downsized by (1/3)^iteration step - 1^. Thus, a structure containing any number of sub-structures can be created. As a result, *D* increases with the number of iteration steps and the related amount of sub-structures (*D*_I1_ = 1.066, *D*_I2_ = 1.113, *D*_I3_ = 1.155, *D*_I4_ = 1.193).

The cross-sectional images of the actual investigated specimens were prepared by separating the structured front face of one hollow cylinder specimen per investigated structure into five radial directions (0°, 22.5°, 45°, 67.5°, 90°). Thereby, anisotropic effects due to preferred directions of the surface structure are considered. Three cross-sectional images were prepared for each cutting direction, resulting in a total number of 15 images per investigated structure. As the fractal dimension approach is independent from scale, the magnification of the cross-sectional images was chosen to achieve a ratio of 0.2 between average maximum structure height and image height. The average maximum structure height was determined using ImageJ 1.52a image evaluation software [22] by measuring the distance between the highest and lowest point of the profile line out of five cross-sectional images for each investigated structure.

### 2.5. Surface Roughness

Tactile surface roughness measurements enable quick and inexpensive characterization of structured surfaces. In this context, a preceding study by Saborowski et al. [10] presented considerable accordance between surface roughness slope tan*θ* and the achieved interlaminar strength for various surface pretreatments in EN AW-6082/PA 6 PMH. Figure 6a illustrates a simplified model of the roughness profile deduced from the work of Chen et al. [15]. Thereby, ideally wedge-shaped profile elements as well as a friction coefficient *µ* between polymer and metal are assumed. The slope angle *θ* represents the relation between distance *RSm* and height *Rz* of the roughness features. In this simplified model, tan*θ* is related to the interlaminar strength in two ways. Firstly, a higher structure density is indicated as more profile elements are arranged within a given distance. Secondly, when a shear force *F*_s_ (e.g., induced by shear load or polymer shrinkage) is applied to the joint, the resulting normal force *F*_n_
*= F*_s_ sin*θ* and the tangential force *F*_t_
*= F*_s_ cos*θ*. An increase in *θ* leads to an increase in *F*_n_. Thus, the maximum friction force *µF*_n_, hindering the polymer from slipping, is increased. *F*_t_, which forces the polymer to slip, is decreased.

It is noteworthy that a certain part of the actual roughness profile is always neglected because of the spatial extension and vertical orientation of the stylus tip. Figure 6b illustrates how tight profile valleys, undercuts, as well as small sub-structures are partially neglected. Thus, tactile roughness measurement can heavily underestimate the actual tan*θ*, especially for small-scaled structures with densely arranged profile elements.

The roughness measurements were carried out using a Hommel-Etamic^®^ T8000 stylus profiler (JENOPTIK AG, Jena, Germany). Five measurements with an evaluation length *l*_n_ of 12.5 mm were recorded and evaluated for each surface structuring process. Thereby, a 2 µm/60° stylus tip was used for capturing the highest possible amount of profile details. The *Rz* values were determined in accordance to ISO 4287, whereas tan*θ* is determined by Equation (5) according to NASA Tech Brief 70-10722. Thereby, *y* is the profile height signal as function of distance *x* within *l*_n_.
(5)$$\tan\theta =\frac{1}{l_{n}}\int_{0}^{l_{n}}\left|\frac{dy}{dx}\right|dx$$

## 3. Results and Discussion

Table 2 summarizes the test results as well as surface characteristics for all investigated surface pretreatments (SD denotes the standard deviation). Figure 7 shows the corresponding scanning electron microscopy (SE) images as well as cross-sectional back scattering detector (BSD) images. The corundum blasted surface (F16) has the highest *Rz* of all investigated surface structures, but second lowest *D*, tan*θ*, and interlaminar strength values. The surface provides several sharp-edged roughness features, but a low number of undercuts. Additional etching (F16-NaOH) leads to the formation of fine dents measuring 5–30 µm in diameter on the coarse blasted structures. However, sharp edges as well as undercuts partially dissolve. As a result, *Rz*, tan*θ*, *D*, and interlaminar strength are slightly lower, even though a higher blasting pressure was used. The thermally sprayed surface (NiAl5) provides higher tan*θ*, *D*, as well as interlaminar strength than the corundum blasted surfaces. Atomization of spraying particles leads to a broad variety of structure sizes, reaching from <1 µm up to around 100 µm. Thereby, conglomeration of spraying particles leads to the formation of undercuts. The deterministic laser structuring processes (L-8, L-11, L-14) lead to pin-like structural elements of 7–10 µm in diameter, whereby a thickening of the pins with the increasing number of scans is observed. The average maximum structure height increases from 27 µm (8 scans) to 53 µm (14 scans). In accordance with the selected pulse and line distance of 13 µm, the distance between the pins is equal for all variations. As a result, *D* increases as expected with the number of scans. The stochastic pin surface structure (L-Pin) is characterized by steep, conical shaped profile elements of 40–80 µm in height and approximately 18 µm in distance. In contrast to the deterministic structures, the profile elements are stochastically distributed. *D* and the interlaminar strength show the highest value of all investigated surface structures.

It is noteworthy that the joined deterministic laser structure specimens partially contained entrapped air in the holes on the metal surface. Any other joined specimen showed complete wetting with polymer. Consequently, an even higher potential of the deterministic laser structures in the case of complete wetting can be assumed, but not proven.

In addition to Table 2, Figure 8 shows the experimental results of all investigated surface pretreatments. Thereby, the shear strength varied in a broad range from 11.5 MPa to 32.5 MPa and the tensile strength from about 2.0 MPa to 26.0 MPa. On closer inspection, it seems obvious that the fractal dimension *D* rather than the parameter tan*θ* is closely correlated to the obtained interlaminar shear and tensile strength. This is further examined in more detail.

As a nonlinear increase of *τ*_max_ and *σ*_max_ is expected with ascending *D* as well as tan*θ*, a second degree polynomial regression was carried out to determine strength prediction functions from the experimental data. Figure 9 shows the individual strength prediction functions for each surface parameter and load case. *τ*_max_ (*D*) shows particularly high accordance (*R*^2^ = 0.99), whereas *σ*_max_ (*D*) shows lower, but still acceptable accordance (*R*^2^ = 0.88). It is noteworthy that an improvement in surface structure affects the tensile strength much stronger than the shear strength. This is also confirmed from the tensile strength/shear strength ratio *σ*_max_ (*τ*_max_) shown in Figure 10, where a slight increase in *τ*_max_ is connected to a strong increase in *σ*_max_. Taking into account the high accordance (*R*^2^ = 0.95), tensile strength prediction from shear strength and vice versa is also reasonable.

In contrast to *D*, tan*θ* fails to predict *τ*_max_ and *σ*_max_ for the laser-structured surfaces. This is related to the densely arranged profile elements that prevent an appropriate roughness measurement because of missing penetration of the stylus tip (Figure 6b). Additionally, the tan*θ* values for the L-8, L-11, and L14 structures show particularly high standard deviations owing to preferred directions of the profile elements. Although preferred directions also increase the standard deviation of the fractal dimension, the effect is significantly lower in all three cases. Consequently, the accordance of the tan*θ*-related strength prediction functions is considerably lower (*τ*_max_ (tan*θ*): *R*^2^ = 0.88, *σ*_max_ (tan*θ*): *R*^2^ = 0.58).

The fractured surfaces of the shear- and tensile-tested specimens are depicted in Figure 11. Dark areas indicate fractured polymer as the images were recorded using the BSD that allows for imaging material contrast. As dark areas also occur in the case of a weak electron signal, the untested surfaces are shown additionally for comparison. In general, an increasing amount of cohesively fractured polymer is observed with increasing interlaminar strength for both load cases. For each surface pretreatment, the shear-tested surfaces contain considerably more remaining polymer than the tensile-tested surfaces. This can be explained by the orientation of the profile elements that predominantly provide undercuts against shear rather than against tensile loads.

The laser structured surfaces are characterized by tight and steep profile valleys that deliver a weak electron signal. Consequently, areas of remaining polymer can hardly be located as most parts of the surface appear dark. To this end, Figure 12 provides a detailed view using the SE-detector. For the shear-tested surfaces, plastically deformed pins are observed for each structure. The amount of fractured polymer increases with the number of scans for the deterministic pin structures. Fractured polymer covers the complete surface on the L-14 as well as on the L-Pin specimens. For the tensile-tested surfaces, only the L-Pin structure shows almost complete cohesive failure.

## 4. Conclusions

On the basis of the experimental results using EN AW-6082/PA6 PMH and the performed analyses in the present work, the following conclusions can be drawn:Interlaminar shear and tensile strength are related to each other. An increase in shear strength corresponds to an increase in tensile strength. Tensile strength prediction from shear strength and vice versa is reasonable for profile element heights within the investigated range (27.5 < *Rz* < 131).Interlaminar strength prediction based on the surface structure is reasonable for profile element heights within the investigated range.Fractal dimension is an appropriate, scale-independent measure for describing the surface structure with a mathematical value as it considers structure density, undercuts, as well as sub-structures.Tactile measured surface roughness slope is an appropriate measure for coarse structures, but fails for undercut, densely arranged, and small-scaled profile elements, as well profile elements arranged in preferred directions.The fracture analysis indicates that the amount of polymer residues on the metal surface is strongly related to the interlaminar strength. Higher interlaminar strength leads to more residues. Shear testing leads to considerably more residues than tensile testing.Wetting behavior of the metal surface with polymer must be considered when predicting the interlaminar strength from the surface structure because incomplete wetting lowers the achievable strength in relation to the theoretical potential.Laser generated, stochastically distributed pin microstructures led to the highest interlaminar strength of all investigated structuring methods. The profile elements were densely arranged, steep, partially undercut, showed complete wetting with polymer, and had the highest fractal dimension.

## Figures and Tables

**Figure 1 materials-13-02171-f001:**
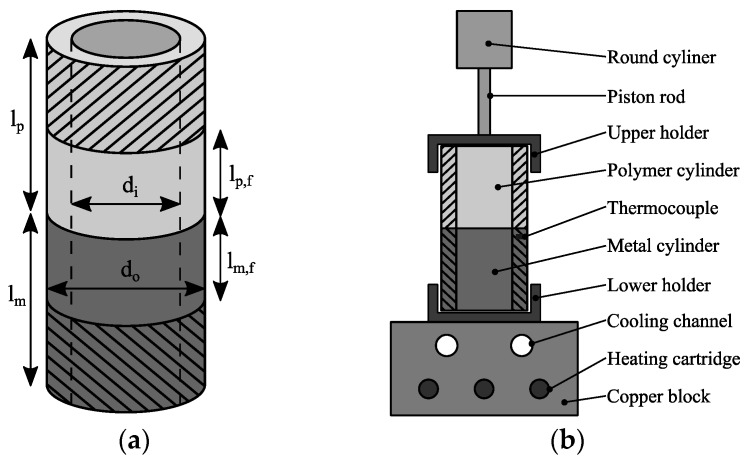
Hollow cylinder specimen: (**a**) geometry and (**b**) hot-pressing tool.

**Figure 2 materials-13-02171-f002:**
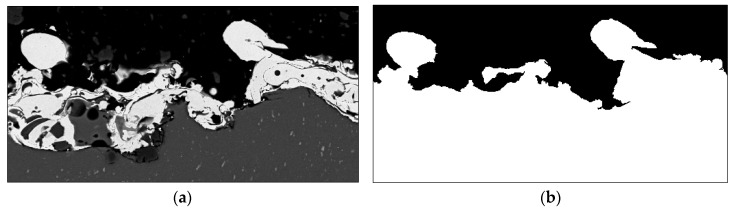
Process of image analysis for determination of fractal dimension: (**a**) cross-section, (**b**) binary image, (**c**) interface line, and (**d**) box-counting algorithm.

**Figure 3 materials-13-02171-f003:**
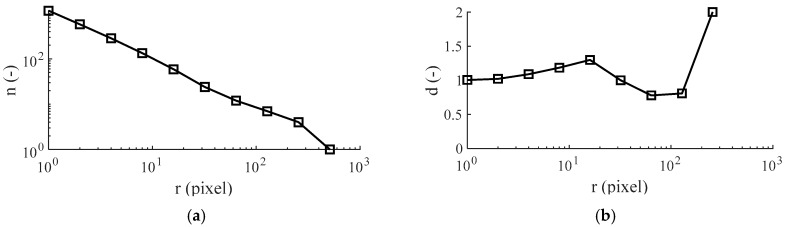
Box-counting: (**a**) number of boxes over box size and (**b**) local dimension over box size.

**Figure 4 materials-13-02171-f004:**
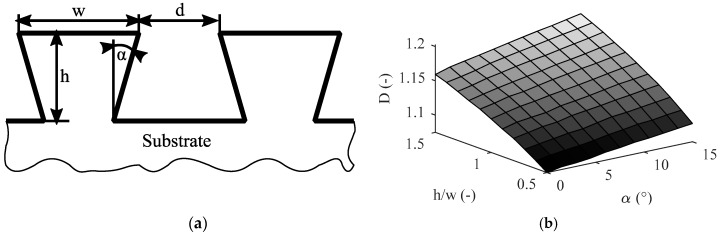
Dovetail structure: (**a**) schematic representation and (**b**) *D* in dependence of undercut angle *α* and height-to-width ration *h/w* (*w/d* = 2.0).

**Figure 5 materials-13-02171-f005:**
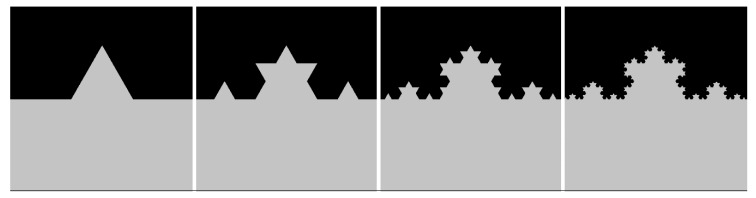
Koch curve, iteration steps 1–4 (from left to right).

**Figure 6 materials-13-02171-f006:**
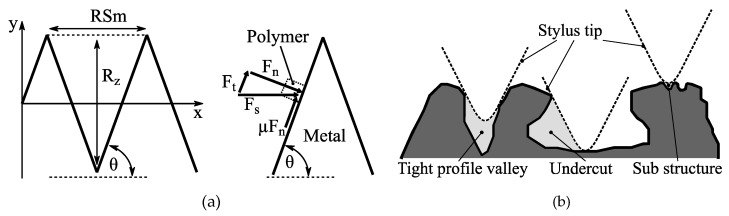
Surface roughness profile: (**a**) schematic representation and resulting forces and (**b**) not considered profile elements.

**Figure 7 materials-13-02171-f007:**
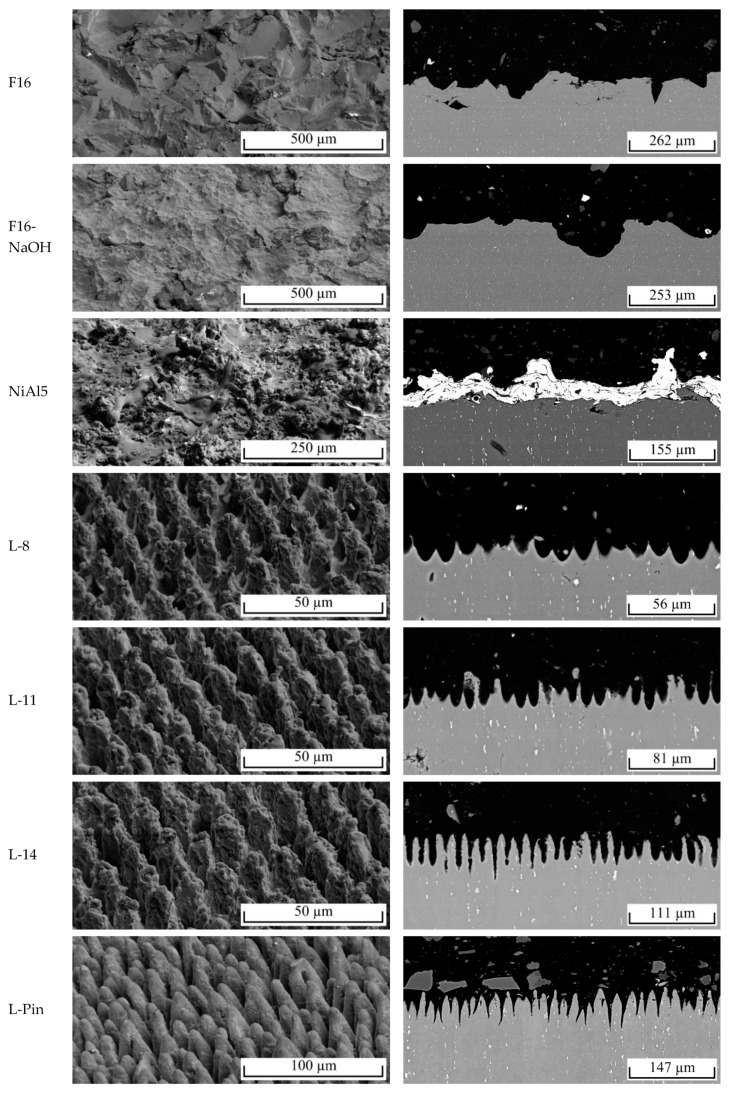
Scanning electron microscopy (SE) images of topography (left) and cross-sectional back scattering detector (BSD) images (right).

**Figure 8 materials-13-02171-f008:**
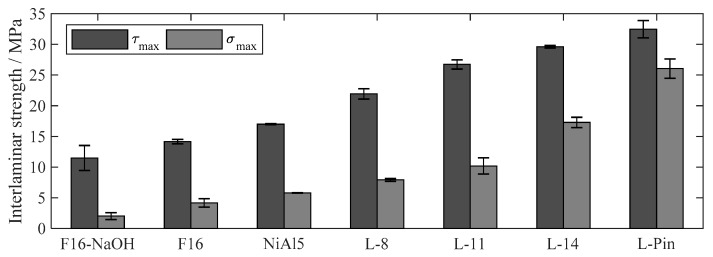
Shear and tensile strength of the hollow cylinder specimens, mean values ± 1 SD.

**Figure 9 materials-13-02171-f009:**
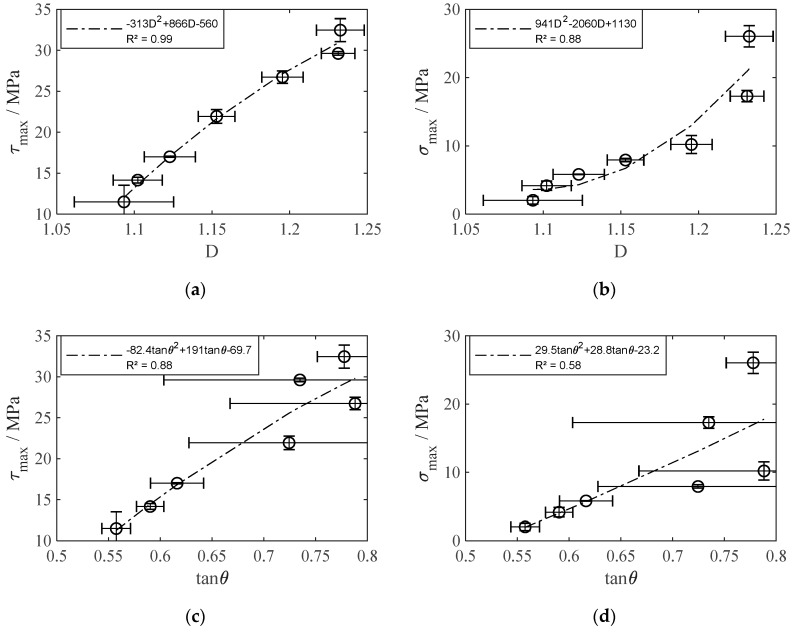
Interlaminar shear and tensile strength in relation to (**a**,**b**) fractal dimension *D* and (**c**,**d**) surface roughness slope tan*θ*; mean values ± 1 SD.

**Figure 10 materials-13-02171-f010:**
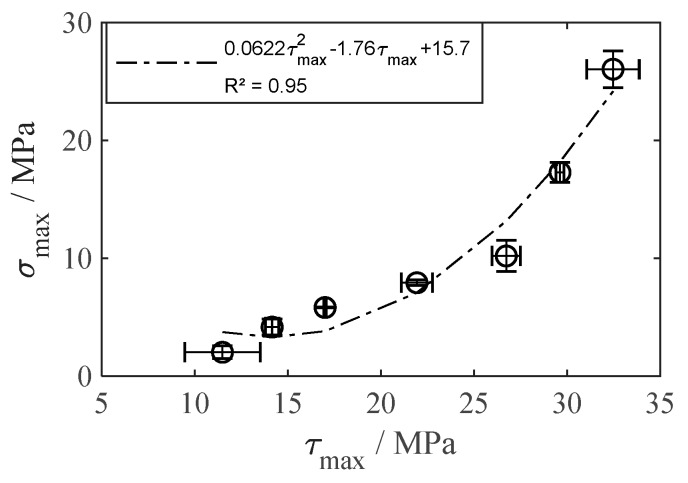
Interlaminar tensile strength *σ*_max_ in relation to shear strength *τ*_max_, mean values ± 1 SD.

**Figure 11 materials-13-02171-f011:**
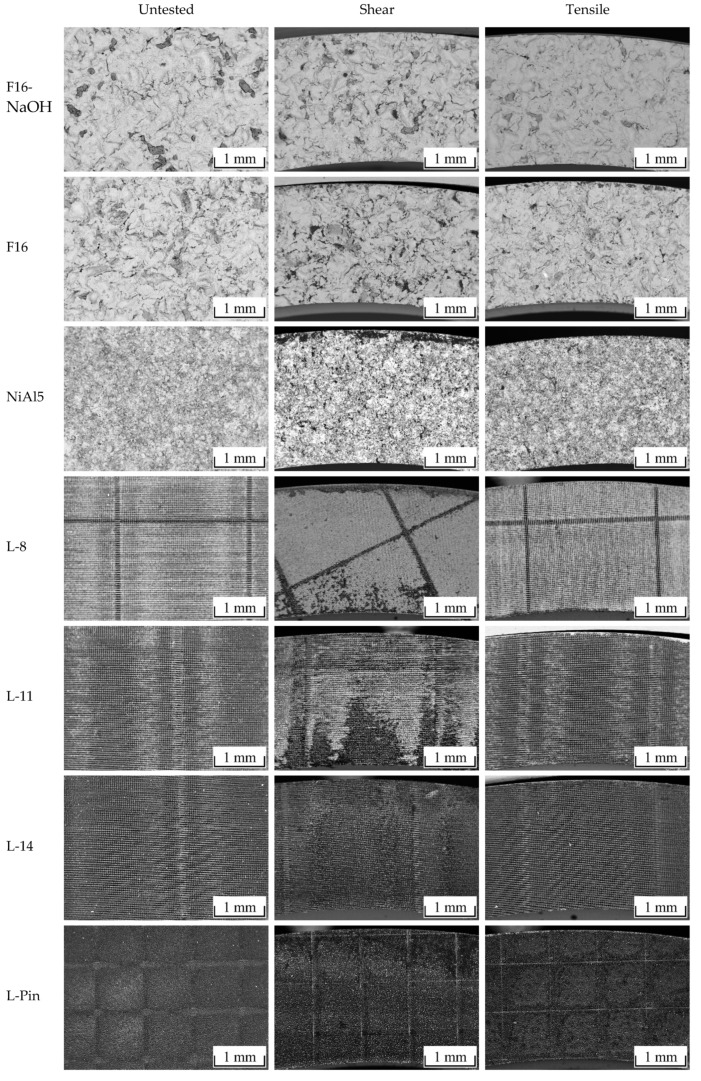
BSD images of the fractured metal surfaces.

**Figure 12 materials-13-02171-f012:**
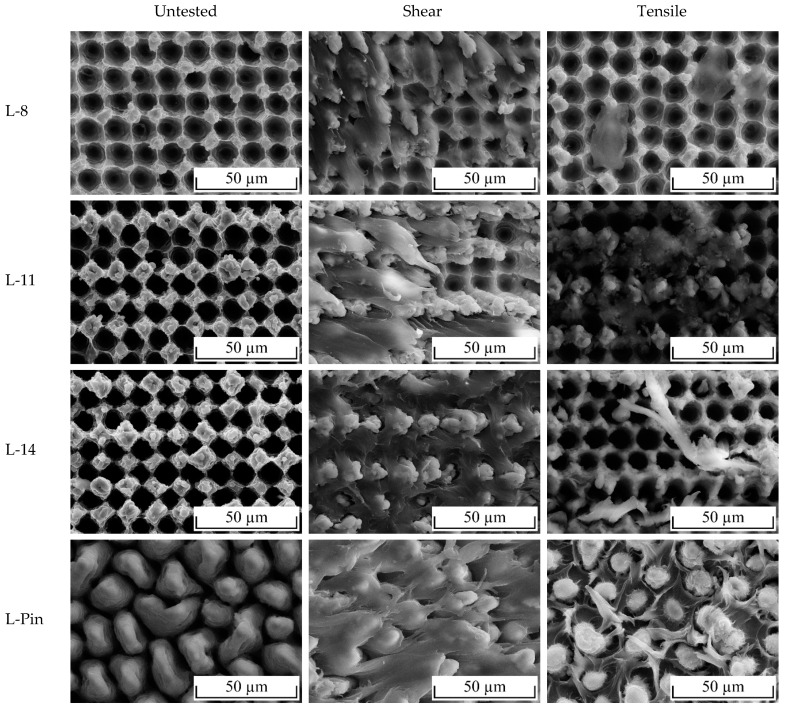
SE images of fractured, laser-structured metal surfaces.

**Table 1 materials-13-02171-t001:** Material properties of the used metal and polymer.

	EN AW-6082	PA 6 (Humid)
Density [kg/m³]	2.7	1.14
Young’s modulus [MPa]	70,000	1800
Poisson’s ratio [—]	0.34	—
Yield strength [MPa]	260	60
Ultimate strength [MPa]	310	—
Elongation to failure [%]	7	200
Melting temperature [K]	933	496
Thermal expansion coefficient [10^−6^/K]	23.4	70
Thermal conductance [W/(m·K)]	170–220	0.23
Specific heat [J/(kg·K)]	898	1700

**Table 2 materials-13-02171-t002:** Roughness parameters, fractal dimension, and interlaminar strength of the investigated surface structuring methods, mean values ± 1 SD.

Treatment	R_z_/µm	tan θ	D	τ_max_/MPa	σ_max_/MPa
F16	131 ± 8	0.590 ± 0.013	1.102 ± 0.016	14.2 ± 0.4	4.16 ± 0.69
F16-NaOH	124 ± 9	0.557 ± 0.014	1.093 ± 0.032	11.5 ± 2.0	2.02 ± 0.55
NiAl5	80.4 ± 4.2	0.616 ± 0.004	1.123 ± 0.016	17.0 ± 0.1	5.81 ± 0.03
L-8	32.2 ± 0.4	0.724 ± 0.025	1.153 ± 0.012	21.9 ± 0.8	7.92 ± 0.24
L-11	31.0 ± 1.1	0.788 ± 0.096	1.195 ± 0.013	26.7 ± 0.8	10.2 ± 1.3
L-14	27.5 ± 2.2	0.735 ± 0.121	1.231 ± 0.011	29.6 ± 0.2	17.3 ± 0.8
L-Pin	45.6 ± 1.2	0.778 ± 0.026	1.233 ± 0.015	32.5 ± 1.4	26.0 ± 1.6

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
