# Peer review of "Introducing Fractal Dimension for Interlaminar Shear and Tensile Strength Assessment of Mechanically Interlocked Polymer–Metal Interfaces"

_materials, 2020, doi:10.3390/ma13092171_

Round 1

Reviewer 1 Report

The paper is in fact a concise report of experiment which compared polymer to metal adhesion when metal surface was prepared in different ways and it showed difference in a surface roughness. The idea is not very original, but results could be useful for engineers develoing combined polymer/metal parts. Using of fractal dimension created a mathematical tool which coud be used for precise comparision of different surfaces.

I have found that in present state paper is ready to print.

Author Response

The paper is in fact a concise report of experiment which compared polymer to metal adhesion when metal surface was prepared in different ways and it showed difference in a surface roughness. The idea is not very original, but results could be useful for engineers develoing combined polymer/metal parts. Using of fractal dimension created a mathematical tool which coud be used for precise comparision of different surfaces.

I have found that in present state paper is ready to print.

Thank you for the review. The presented structuring methods are indeed not new, but such a good correlation between a surface measure and the interlaminar strength is. The idea behind it is to be able to perform a surface design based on a scalar measure in addition to the pure surface characterization. This is to be demonstrated in a future research project using a turning process.

Reviewer 2 Report

The reviewed paper is aimed to the determination of most representative parameter to characterize the shear and tensile strength of the metallic surface of the structures with mechanically interlocked polymer-metal-interfaces. The present material contains many experimental results that are very useful for the practice of the lightweight structure design and manufacturing. It should be noted the detailed description of the experimental procedure including the specimens preparation that is very important for the engineering practice as well as for the experimental studies of another investigators. The paper is well structured, the presentation is logical and done in a good language. This article can be recommended for publication without a doubt.

Some suggestions to the authors.

Line 244. It is advisable to refer to ImageJ image evaluation software developer.

Subsection 2.2 describes in detail the surface pretreatment methods used, but the first column of Table 2 contains abbreviations for these methods. This creates difficulties for readers because they must return to descriptions 2.2. I recommend inserting these abbreviations into the descriptions of the preprocessing methods in 2.2.

Figure 8 shows the values of the interlayer shear and tensile strength for samples processed by various methods. Each solid column (average strength value) is accompanied by a confidence interval. It is desirable to present the value of confidence probability, which was used in calculating the boundaries of these intervals. Moreover, all the graphs presented in Figure 9 will be much more convincing if confidence intervals are presented on the graphs along with dots.

I think that the conclusions may contain some recommendations for choosing the best technology for pre-treatment of bonded metal and polymer surfaces.

Author Response

Thank you very much for the review and the positive comments. We have implemented their suggestions as follows.

Line 244. It is advisable to refer to ImageJ image evaluation software developer.

A reference as desired by the software developers has been added.

Subsection 2.2 describes in detail the surface pretreatment methods used, but the first column of Table 2 contains abbreviations for these methods. This creates difficulties for readers because they must return to descriptions 2.2. I recommend inserting these abbreviations into the descriptions of the preprocessing methods in 2.2.

The abbreviations have been introduced in chapter 2.2.

Figure 8 shows the values of the interlayer shear and tensile strength for samples processed by various methods. Each solid column (average strength value) is accompanied by a confidence interval. It is desirable to present the value of confidence probability, which was used in calculating the boundaries of these intervals. Moreover, all the graphs presented in Figure 9 will be much more convincing if confidence intervals are presented on the graphs along with dots.

The confidence interval is +- 1 standard deviation (~68% confidence probability if normal distribution is assumed). A comment to that has been made in every caption. Additionaly, the confidence intervals have been added to Figure 9. Some additional comments to that were made within the text.

I think that the conclusions may contain some recommendations for choosing the best technology for pre-treatment of bonded metal and polymer surfaces.

A short description of the structuring method providing the highest strength was added.

Reviewer 3 Report

Saborowski and coworkers utilized metal surface dimension for evaluating the interlaminar shear and tensile strength of polymer-metal-interfaces. The experiments were carried out on a butt-bonded hollow cylinder method by which the shear and tensile strength can be determined within one specific specimen geometry. The finding indicated that the fractal dimension is a better measure for predicting the interlaminar strength in comparison with the surface roughness. The manuscript is well written. Therefore, I would recommend it to be published in Materials. One minor comment is provided for the authors' consideration. Chemical compositions of polymers as well as the surface properties of metal alloys may also have an influence on the interlaminar strength. Could the authors give some comments on the generality of this fractal dimension evaluation method for different combinations of polymers and metal surfaces?    

Author Response

Thank you very much for your review and the positive comments.

Chemical compositions of polymers as well as the surface properties of metal alloys may also have an influence on the interlaminar strength. Could the authors give some comments on the generality of this fractal dimension evaluation method for different combinations of polymers and metal surfaces?

In fact, not all polymer-metal composites can be concluded from one material combination. In the future we will investigate further material combinations (including polypropylene) and modify the surface chemistry with silanes. However, the results are not yet available, so unfortunately we cannot yet make any scientific statement on this.

Reviewer 4 Report

This paper describes polymer-metal hybrid material and their surface adsorption using analysis of fractral.

Although wet drop has been investigated using such method, polymer-metal adsorption is novel. So it must be a significant paper in Materials.

No serious problems could be found, and is should be accepted as it is. 

If possible, please let them add references furthermore.

Author Response

Thank you very much for the review.